# Effects of Water Immersion Versus Epidural as Analgesic Methods during Labor among Low-Risk Women: A 10-Year Retrospective Cohort Study

**DOI:** 10.3390/healthcare12191919

**Published:** 2024-09-25

**Authors:** Carmen Herrero-Orenga, Laura Galiana, Noemí Sansó, Myriam Molas Martín, Araceli Castro Romero, Juan Carlos Fernández-Domínguez

**Affiliations:** 1Obstetrics and Gynecology Service, Maternal Ward Unit, Hospital of Inca, 07300 Inca, Spain; mariac.herrero@hcin.es (C.H.-O.); myriam.molas@hcin.es (M.M.M.); araceli.castro@hcin.es (A.C.R.); 2Department of Nursing and Physiotherapy, University of Balearic Islands, 07122 Palma, Spain; jcarlos.fernandez@uib.es; 3Health Research Institute of the Balearic Islands (IdISBa), 07120 Palma, Spain; 4Department of Methodology for the Behavioral Sciences, University of Valencia, 46010 Valencia, Spain; laura.galiana@uv.es

**Keywords:** water immersion, waterbirth, epidural, analgesia, pain relief, maternal and newborn outcomes, perineal trauma

## Abstract

Background: Adequate pain relief during childbirth is a very important issue for women and healthcare providers. This study investigates the effects on maternal and neonatal outcomes of two analgesic methods during labor: water immersion and epidural analgesia. Methods: In this retrospective observational cohort study at a first-level hospital, in Spain, from 2009 to 2019, 1134 women, low-risk singleton and at term pregnancy, were selected. Among them, 567 women used water immersion; 567 women used epidural analgesia for pain control. Maternal outcomes included mode of birth and perineum condition. Neonatal outcomes included 5 min Apgar score, umbilical cord arterial pH, and Neonatal Intensive Care Unit admissions. Chi-square tests and Mann–Whitney *U* tests, together with their effect sizes (Cramer’s *V*, odds ratio, and Cohen’s *d*) were used to test the main hypotheses. Results: Spontaneous vaginal birth was almost 17 times more likely in the water immersion group (OR = 16.866 [6.540, 43.480], *p* < 0.001), whereas the odds of having a cesarean birth were almost 40 times higher in the epidural group (OR = 39.346 [3.610, 429.120], *p* < 0.001). The odds of having an intact perineum were more than two times higher for the water immersion group (OR = 2.606 [1.290, 5.250], *p* = 0.007), whereas having an episiotomy was more than eight times more likely for the epidural group (OR = 8.307 [2.800, 24.610], *p* < 0.001). Newborns in the water immersion group showed a better 5 min Apgar score and umbilical cord arterial pH and lower rates in admissions at the Neonatal Intensive Care Unit. Conclusions: Women choosing water immersion as an analgesic method were no more likely to experience adverse outcomes and presented better results than women choosing epidural analgesia.

## 1. Introduction

The presence of pain during childbirth is one of the most important causes of stress and worry for pregnant women. Its multifactorial origin, as a combination of physiological and biobehavioral factors, leads to great variability in women’s perception [1]. 

For women at low risk of childbirth complications, who desire less invasive, non-pharmacological pain relief options, that respect the physiology of childbirth, water immersion (WI) is an analgesic option that has gradually increased in popularity and acceptance in most high-income countries [2,3]. Beyond pain relief, multiple other benefits of WI during labor have been reported in the literature [2,3,4,5,6,7]. Nonetheless, its utilization rates vary widely by country [4,8], and its use remains limited and affected by medically led environments that restrict pool use [9]. 

WI during labor has been shown to reduce the perception of painful contractions [10,11,12], unnecessary interventions [4,13,14], and the use of intrapartum pharmacological analgesics [2,4,11,15] while increasing maternal relaxation [11], satisfaction, empowerment, and active participation in decision-making [11,16,17].

In addition, WI seems to improve obstetric results, following rigorous selection criteria [5,6,7], without increasing the risks or adverse maternal and/or neonatal effects [2,4,8,14,18,19,20,21]. Some studies show that WI increases the incidence of spontaneous vaginal births (SVBs) and decreases rates of operative vaginal births (OVBs), vacuum cups, or forceps [2,22]. Regarding the rate of cesarean births, there are contradictory results [4,21,22]. Although some studies have related WI to a higher rate of severe perineal lacerations [23,24], most studies refer to an increased rate of intact perineums [8,13,18], lower rates of minor perineal lacerations [3,8,14,18,25,26], and especially a decrease in severe perineal tears [8,26] and episiotomies [2,3,8,12,27,28,29]. 

Regarding neonatal outcomes, there is still some controversy. Some studies confirm that there are no significant differences in rates of neonatal infections [21,30,31,32], Apgar score [14,18,21,25,27,29,30,32], postpartum umbilical cord pH [4], and rates of admission to Neonatal Intensive Care Units (NICUs) [8,14,18,19,26,29,30,32]. WI has not been associated with a greater need for cardiopulmonary resuscitation at birth, greater frequency of respiratory syndromes, or worse neonatal outcomes in general [4,10,12,30,31] when comparing outcomes between conventional birth and WI. Nevertheless, and although statistical significance was not always achieved, some studies present favorable results for WI in terms of fewer non-pneumonia infections [31], fewer respiratory complications [12,31], higher Apgar score at 5 min [30], and lower rates of admission to NICU. [12,25,31]. On the other hand, other researchers suggest that further studies are needed on the safety of WI for neonates during the second stage of birth [33,34]. 

Currently, epidural analgesia (EA) is considered the first choice of pharmacological analgesic method in obstetric, due to its high effectiveness in reducing pain, superior to any other non-pharmacological methods [35]. EA allows a woman to remain alert during labor and avoid some systemic side effects of analgesic drugs on the baby. A functioning EA allows the option of regional anesthesia for interventions such as cesarean section or manual removal of a retained placenta, thereby avoiding the risks associated with general anesthesia. However, it is known that EA can carry some risks such as prolongation of labor stages, maternal hypotension, motor blockage, maternal fever, and urinary retention [35]. 

Regardless of the chosen method of pain relief, it is imperative that it be effective and safe for the mother and the newborn [35].

Most literature reviews focus on studies comparing WI with conventional birth without analgesia; however, today there are many more women who use EA in their birth process than those who choose not to use any kind of analgesia. Therefore, there is a need to deepen the knowledge of the differences between childbirth in which pharmacological methods of pain relief are used, specifically, EA, and the use of WI. This knowledge will probably help to clarify the existing controversies in some obstetric and neonatal results in relation to the use of both pain relief methods.

Our goal is to study the differential effect between two analgesic methods for pain relief in childbirth, WI versus EA, in maternal and neonatal outcomes. Two main hypotheses were tested: Women in the WI group will present better obstetric results when compared to women in the EA group:▪Regarding the mode of birth, women in the WI group will present higher SVB rates, whereas the EA group will present higher cesarean rates.▪Regarding perineal integrity, higher rates of intact perineum will be found in the WI group, whereas the EA group will present higher episiotomy rates.WI will not pose a health risk to newborns when compared to the EA group:▪Newborn 5 min Apgar score and umbilical cord arterial pH in the WI group will not present statistically significant differences when compared to those in the EA group.▪Newborns facing challenges in transitioning to life outside the womb rates will not differ between groups.▪NICU admissions will not differ between groups.

## 2. Materials and Methods

A retrospective observational cohort study was carried out to analyze the differential effect between two analgesic methods for pain relief in childbirth: WI versus EA. The Strengthening the Reporting of Observational Studies in Epidemiology recommendations were considered in the report of this study. 

### 2.1. Research Setting

The research setting was a Spanish first-level public hospital, completely funded by the National Health System. An average of 980 births per year were attended to during the study period. Approximately 9% used WI as an analgesic method, and 45% used EA. 

### 2.2. Recruitment and Inclusion/Exclusion Criteria

Women, upon admission, could choose the analgesic method according to their preferences and were screened prior to starting WI or EA. The screening included the confirmation that maternal vital signs were normal, gestational age was between 37 and 42 weeks, low-risk pregnancy criteria were met, and there was a singleton fetus in cephalic presentation. In addition, the women had to be in active labor, and they should not require stimulation with drugs for the onset of birth. A 20 to 30 min Electronic Fetal Monitoring had to be performed before the analgesic method was started, and the trace result had to be category 1. See Appendix A for exclusion criteria.

All women in both groups had pregnancy check-ups by midwives and obtained similar information about the birth process, available analgesic methods, the importance of diet and physical exercise during pregnancy, antepartum perineal massage, and pelvic floor exercises.

### 2.3. Participants

In total, 567 women were included in the study as the WI group. They were all the women who chose immersion in water as an analgesic method during childbirth from June 1st 2009 to May 31st 2019 and began WI after admission.

Regarding the EA group, 4445 women chose epidural as an analgesic method for childbirth during the study period, despite meeting the conditions to use WI, and met the inclusion criteria. Out of these 4445 eligible participants, 567 women were selected following a stratified random sample 1:1. The sample was stratified by parity. To randomize the sample, the Excel formula (RANDOM) was used to assign a random number to each row of the list. Then, the number of records needed for the study was indicated (the same as the number of women who had used water immersion in each sample stratum). 

In all, 1134 women, who met the clinical criteria, were included in the study: 567 women for the WI group and 567 women for the EA group, as shown in Figure 1.

### 2.4. Main Outcome Measures

Measures for group comparability included parity, women’s age, and newborns’ weight. 

Obstetric outcomes included mode of birth and perineum condition. Birth was categorized into SVB and OVB, which included any birth that ends with the application of a suction cup or forceps and cesarean birth. Perineum condition included intact perineum, minor lacerations (first- or second-degree tears), major lacerations (third- or fourth-degree tears), and episiotomy.

Finally, newborn outcomes included 5 min Apgar score, umbilical cord arterial pH, and NICU admissions. Five-minute Apgar score and umbilical cord arterial pH were used to classify newborns with difficulties in the transitional period (NDTPs) and healthy newborns (HNs), with those who presented an Apgar score < 7 at 5 min of life and/or a postpartum umbilical cord pH < 7.10 being defined as NDTPs, as defined in the literature [36,37]. At the study center, postpartum umbilical cord pH is routinely measured. The sample is taken after the first 60 s of life and is immediately analyzed in a gasometer located in the delivery room. 

### 2.5. Statistical Methods

The data collection was carried out by computer, from the maternal ward dashboard and the electronic medical records of the women. Baseline characteristics of each group were compared in terms of percentage of parity, women’s age, and newborns’ weight. The exact chi-square statistic, Kolmogorov–Smirnov test, and Mann–Whitney *U* test, with associated effect sizes, were used. 

To test the first hypothesis, 2 chi-square tests were performed together with their association measures. A chi-square test compared the mode of birth across groups (SVB, OVB, and cesarean birth). In the second chi-square test, women were distributed into intact perineum, minor lacerations, major lacerations, and episiotomy across groups. The standardized residuals were used to explore in which categories the frequencies observed were different from the frequencies expected. Regarding the effect size, Cramer’s *V* was calculated. Additionally, the marginal odds ratio (for 2 × 3 and 2 × 4 tables) and 95% interval of confidence were calculated for each category. 

To test the second hypothesis, newborns’ scores from the WI and EA groups were compared in terms of 5 min Apgar score and umbilical cord arterial pH, using Kolmogorov–Smirnov and Mann–Whitney *U* tests and Cohen’s *d*. Two chi-square tests were used to study the frequency distribution of NDTPs across the water immersion and the epidural groups and the distribution of NICU admissions across the groups under study, respectively. In both cases, standardized residuals, Cramer’s *V*, and the odds ratio (for 2 × 2 tables) and 95% interval of confidence for each category were also calculated. As for neonate NICU admissions, no cases were found in the WI group; the Haldane–Anscombe correction was applied. 

Analysis was performed using SPSS version 26 and R studio, employing the package ggplot2. 

## 3. Results

A total of 1134 women participated in the study (WI, n = 567; EA, n = 567). The sample characteristics of study groups are shown in Table 1. The women in the two groups were similar regarding parity (*χ*^2^(4), 0.38; *p* = 0.98, Cramer’s *V*, 0.02). The mean age was 29.83 (standard deviation = 5.51) years. There were statistically significant differences in the distribution of age across groups, although of small magnitude (*χ*^2^(3), 11.14; *p* = 0.01, Cramer’s *V*, 0.10). More women in the EA group were in the younger groups (25 years old or younger and 26–30 years old), whereas more women in the WI group were older (groups of 31–35 years old and 36 years old or older showed higher rates). Finally, newborns’ weight was compared across groups. As the variable showed statistically significant differences with the normal distribution (*D*(1130), 0.03; *p* = 0.02), the Mann–Whitney *U* test was used. The test showed statistically significant differences (*U*(*n_epidural_* = 563, *n_water immersion_* = 567), 148,681.00; *z*, −1.99; *p* = 0.04, Cohen’s *d*, 0.12), with higher weight in the EA group. However, the mean neonatal weight at birth did not exceed 3500 g in any of the groups. 

Table 2 shows the comparison across the two groups with respect to mode of birth and perineal integrity. Concerning mode of birth, there were statistically significant differences between WI and EA groups (*χ*^2^(2), 278.25; *p* < 0.001; Cramer’s *V*, 0.50). The WI group had the highest rates of SVB (n = 536 [94.5%]), whereas the EA group showed the highest rates of OVB (n = 155 [27.5%]), and cesarean birth (n = 123 [21.8%]). Regarding the odds, if women used WI, the odds of having an SVB were almost 17 times higher than if they used EA (OR, 16.87; 95%CI, 6.54–43.48; *p* < 0.001), whereas if women used EA, the odds of having an OVB were more than 7 times greater (OR, 7.60; 95%CI, 2.74–21.06; *p* < 0.001), and the odds of having a cesarean birth were almost 40 times higher (OR, 39.35; 95%CI, 3.61–429.12; *p* < 0.001). Standardized residuals, displayed in Figure 2, pointed to statistically significant differences in the three categories or modes of births.

Women who had a cesarean birth were excluded from the analysis of perineal integrity. When both groups were compared, there were also statistically significant differences between the WI and EA groups (*χ*^2^(3), 117.57; *p* < 0.001, Cramer’s *V*, 0.34). The WI group had the highest rates of an intact perineum (n = 173 [30.7%]) and minor lacerations (n = 365 [64.8%]), whereas the EA group showed the highest rates of major lacerations (n = 4 [0.9%]) and episiotomy (n = 115 [26.1%]). If women used WI as the analgesic method, the odds of having an intact perineum were more than twice higher (OR, 2.61; 95%CI, 1.29–5.25; *p* = 0.007). However, using WI did not affect the odds of having minor lacerations when compared to the EA group (OR, 1.31; 95%CI, 0.74–2.32; *p* = 0.16), and similarly, using EA did not affect the odds of having major lacerations when compared to the WI group (OR, 2.57; 95%CI, 0.05–125.62; *p* > 0.99). The analgesic method indeed affected the odds of having an episiotomy, which were more than eight times higher for the EA group (OR, 8.31; 95%CI, 2.80–24.61; *p* < 0.001) when compared to the WI method. Standardized residuals, displayed in Figure 2, pointed to statistically significant differences only in the categories of intact perineum and episiotomy.

Finally, comparisons of newborn outcomes across the two study groups are shown in Table 3. As both the 5 min Apgar score (*D*(1133), 0.49; *p* < 0.001) and the umbilical cord arterial pH (*D*(1107), 0.04; *p* < 0.011) showed statistically significant differences with the normal distribution, Mann–Whitney *U* tests were used. The results pointed to statistically significant differences across groups in 5 min Apgar score (*U*(*n_epidural_* = 566, *n_water immersion_* = 567), 138,235.50; *z*, −6.80; *p* < 0.001; Cohen’s *d*, 0.24) and in umbilical cord arterial pH (*U*(*n_epidural_* = 549, *n_water immersion_* = 558), 131,812.00; *z*, −4.02; *p* < 0.001; Cohen’s *d*, 0.24). In both cases, differences favored the WI group, which showed higher scores, with small effect sizes. 

When newborns were categorized into NDTPs and HNs, there were statistically significant differences between WI and EA women (*χ*^2^(1), 7.31; *p* = 0.007; Cramer’s *V*, 0.08). The EA group had the highest rate of NDTPs (n = 29 [5.1%]), and the WI group had the highest rate of HNs (n = 555 [97.9%]). That is, if women used EA, the odds of having an NDTP were more than two times higher (OR, 2.49; 95%CI, 1.26–4.94; *p* = 0.004). Standardized residuals are displayed in Figure 3. 

Outcomes for neonates admitted into the NICU were also different across the groups under study (*χ*^2^(1), 19.32; *p* < 0.001; Cramer’s *V*, 0.13). Newborns in the EA group had the highest rate of NICU admissions (n = 19 [3.4%]), and the WI group had the highest rate of non-admissions (n = 567 [100.0%]). So, if women used EA as the analgesic method, the odds of NICU admissions were 40 times higher than if they used WI (OR, 40.35; 95%CI, 2.43–669.94; *p* < 0.001). Standardized residuals pointed to statistically significant differences in the category of NICU admissions (see Figure 3).

## 4. Discussion

This study analyzed the differential effect between two analgesic methods for pain relief in childbirth: WI versus EA. For this purpose, obstetric and neonatal outcomes were compared in both groups of women.

Concerning baseline characteristics, it should be noted that although the groups did differ in terms of women’s age and newborns’ weight, the differences observed were of negligible size.

As shown in the present study, WI increases the rate of SVB and decreases the rate of OVB and cesarean births when compared to EA use. Former studies comparing WI with conventional births (i.e., without analgesic methods) could not demonstrate significant differences regarding the mode of birth [38,39]. However, recent studies have shown remarkable differences, like ours. Liu et al. [21] stated that women who used WI presented lower cesarean birth rates than women with conventional births (32.9% vs. 13.2%, *p* = 0.026), although data on OVB were not provided. Likewise, Henderson et al. concluded that 95.6% of women who used WI had an SVB [2]. The SVB rates are similar in our study. In 2020, Maude and Kim concluded that SVBs are more frequent in the WI group, reaching rates of 84%, while the New Zealand national average SVB rate was 65.2%, and the rate of OVB was 9.4% and that of cesarean births was 6.5% [22]. Also, Barry et al. found an SVB rate of 84.7% in WI women versus 72.6% in a standard care group. Women in the WI group were less likely to require an OVB (11.1% vs. 24.7%). However, the cesarean birth rate was 4.2% in the WI group compared to 2.6% in the control group [19]. It is known that a cesarean section can lead to the late development of multiple morbidities in both childhood and adulthood, such as obesity at an early age, food allergy, asthma, type 1 diabetes mellitus, and various forms of dermatitis [40], also being a potential risk factor for fetal/neonatal morbidity in subsequent pregnancies, with the possible occurrence of preterm labor, unexplained stillbirth, placenta previa, invasive placenta accreta, and even impacted fetal head [41]. On the other hand, OVBs have been associated with an increased incidence of occult anal sphincter injury and late urinary and anal incontinence in women [42]. 

We have found significant differences in terms of perineal trauma between both groups, with severe perineal trauma being less frequent in the WI group. It should be mentioned that the hospital systematically follows the recommendations regarding the duration of the second stage of labor established by the Clinical Guidelines of Labor Care of the Spanish Ministry of Health, in accordance with the WHO recommendations. It is allowed to increase the duration of the second stage of labor, by one hour, in women who choose EA, since it is known that its use lengthens this stage [35]. The passive phase of the second stage of labor is respected [43], and Valsalva or open-glottis pushing is allowed, based on women’s preference, due to there being no differences in maternal and perineal outcomes between both types of pushing, except for a longer duration of the second stage with open-glottis pushing [44], an outcome that has not been measured in this study. Therefore, both the duration of the second stage of labor and the type of pushing are respected in both groups and therefore do not imply a bias in terms of the results of perineal trauma or mode of birth.

Mollamahmutoglu et al. analyzed outcomes on perineal trauma comparing WI with EA, concluding that the WI group had lower rates of episiotomy but more perineal tears, with no tear degree specification [29]. Other studies compared WI with conventional birth without EA. Snapp et al. found an association between WI and a significant decrease in the episiotomy rate (RR = 0.068; 95%CI, 0.04–0.12), with an absolute risk reduction of 1.8%, and in the likelihood of suffering a tear (RR = 0.98; 95%CI, 0.97–0.99), with an absolute risk reduction of 1.4% [12]. Likewise, the literature review by Nutter et al. states that most studies have linked WI not only with a decrease in the rate of episiotomies, but also with a lower incidence of severe perineal tears and more intact perineums [8], as is also shown in our study. Garland et al. evaluated the rate of severe perineal tears by parity, concluding that primiparous women had the same probability of suffering a third-degree tear, while multiparous women experienced fewer third-degree tears when they used WI (0.2%) compared with conventional birth (0.6%) [45]. Although we have not included the medium/long-term pelvic floor results in the current study, we could assume that the decrease in severe perineal trauma and episiotomy rates associated with WI should lead to fewer alterations in the pelvic floor muscles, being beneficial for the preservation of pelvic floor musculature and strength at 6–8 weeks postpartum [28]. In particular, the restrictive use of mediolateral episiotomy could reduce the incidence of sexual dysfunction, with decreased libido, decreased ability to achieve orgasm, decreased lubrication, disruption of the vaginal epithelium, and the development of vaginal atrophy that can cause dyspareunia [46].

Regarding the Apgar score, some investigations claim that 1 and 5 min Apgar scores are higher in the newborns of women who chose WI, although without significant differences compared to conventional birth [18,25,30,32]. Jacoby et al. reported that only 0.2% of water newborns showed an Apgar score of less than 7 at 5 min, compared to 0.8% in conventional birth (*p* = 0.006) [26]. Also, in agreement with our results, in 2021, Uzunlar et al. demonstrated that 1 and 5 min Apgar scores were significantly lower in women who used EA when they were compared with WI during the first stage of labor [47].

Nutter et al. concluded that, although there are no significant differences in newborn blood gases and pH, WI is associated with similar or improved umbilical cord gases [8]. Our results indicate that there are significant differences in the neonates’ umbilical cord arterial pH, in favor of WI, as reported by Geissbühler and Eberhard, who compared WI with conventional births (7.30 versus 7, 26 *p* < 0.0001) [48].

Although our study did not include neonatal infection rates as an outcome, the literature does not show significant differences in this variable [8,25,31]. However, our research did determine the rate of NDTPs as a main outcome, showing that it is lower in the WI group, with statistically significant differences between the two studied groups, endorsing the results obtained by Vanderlaan et al. [31].

The literature reports that neonatal mortality rates, both in WI and in conventional births, are low [8], and there does not appear to be a direct relationship between WI and adverse neonatal outcomes [8,10,11,18,31]. We could also claim that our research did not find any serious adverse events in either of the two groups.

Most studies report that there are no significant differences in NICU admission rates [4,8,18]. A recent study by Sidebottom et al. in 2020 states that NICU admissions were lower in the WI group during the first and second stages of labor and that there were no significant differences between women who only used WI in the first stage of labor and women who underwent conventional birth [25]. Another study comparing maternal and neonatal outcomes of WI with conventional birth concluded that newborns in the WI group had fewer respiratory complications, NICU admissions, and transfers to tertiary hospitals [12].

Studies comparing WI with EA birth are conflicting regarding NICU admission rates. Mollamahmutglu et al. stated that there were no significant differences and no deaths or infections occurred in either group [29]. However, Uzunlar found that EA increases NICU admission rates compared to WI and SVB without analgesic drugs [47]. Our research also reflects lower rates of NICU admissions in the WI group.

Finally, it is noteworthy that despite the women in the EA group being younger, obstetric outcomes were better in the WI group. This seems to strengthen the results of the present study since advanced maternal ages are related to worse obstetric outcomes [49].

### Study Limitations

The present study has several shortcomings. Firstly, this is a retrospective study, and therefore, the data recording was carried out in the computerized clinical history by different healthcare providers. However, a negligible number of missing data was found in the extracted data. Moreover, the sample size was large and obtained over a period of 10 years. Secondly, the adoption of different maternal positions and mobility during labor could lead to differences in some obstetric outcomes. WI allows free movement and vertical positions, unlike EA in which the adoption of different positions and movements is clearly limited. Free movement and upright positions have been associated with better obstetric outcomes without negative effects on the mother or newborn [50]. However, it seems that the maternal position in women who used EA does not modify birth outcomes [51]. Furthermore, a recent systematic review states that the main difference between upright and supine positions is the shortening of the duration of birth [50], which has not been measured in this study. Finally, there are several confounding variables that could have affected the results of the study. Examples include the influence of physical activity during pregnancy and the practice of antenatal perineal massage. In this sense, Michaleck et al. report that physical activity is safe during pregnancy, but that there are no significant differences in birth outcomes [52]. Only Domenjoz et al. reported a lower risk of cesarean birth in women who carried out a structured program of physical activity during pregnancy [53]. On the other hand, Abdelhakim et al. reported that antenatal perineal massage reduces the risk of severe tears and episiotomies [54]. These variables, which may affect various obstetric and/or neonatal outcomes, were not measured, as there was not a specific registry on these main outcomes, and therefore, this retrospective information could not be gathered. However, we would like to note that all women equally received advice about physical activity during pregnancy and antepartum perineal massage.

## 5. Conclusions

This study demonstrates that women choosing WI as an analgesic method were no more likely to experience adverse outcomes than women choosing EA.

WI is a valuable form of pain relief for women in labor with many benefits. Nowadays, many women still do not have access to this analgesic method. EA rates are high and associated with both positive and negative outcomes. To enable better-informed decision-making about women’s analgesic options in childbirth and to better inform policies and clinical guidelines, evidence regarding the outcomes of WI is much needed. A comparison of WI outcomes to EA outcomes is a valuable contribution, and the knowledge and subsequent disclosure of these results will allow healthcare providers to offer objective, truthful, and precise information to guide women in making decisions.

## Figures and Tables

**Figure 1 healthcare-12-01919-f001:**
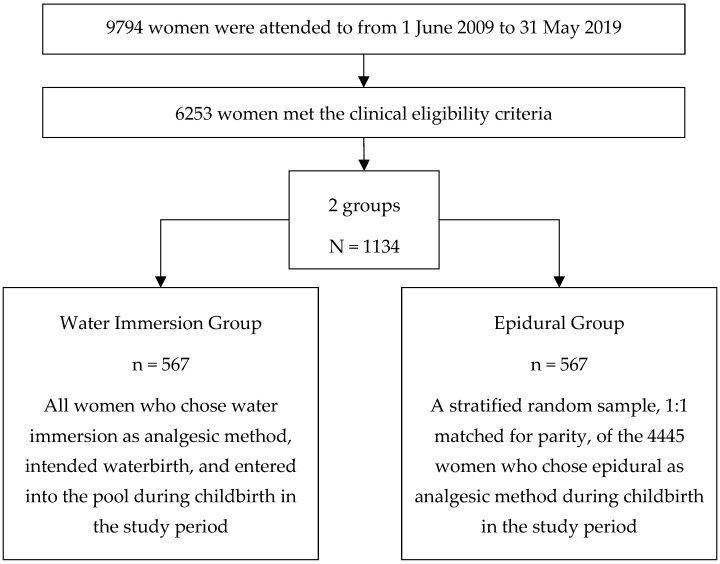
Women attended to during the study period at the maternity ward and establishment of the groups.

**Figure 2 healthcare-12-01919-f002:**
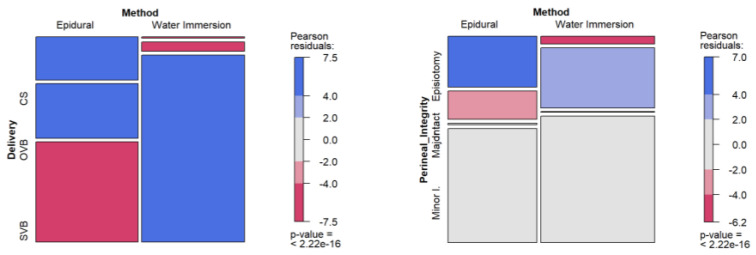
Standardized residuals for maternal outcomes among study groups. Notes: SVB = spontaneous vaginal Birth; OVB = operative vaginal birth; CS = cesarean section; Minor l. = minor laceration; Major l. = major laceration.

**Figure 3 healthcare-12-01919-f003:**
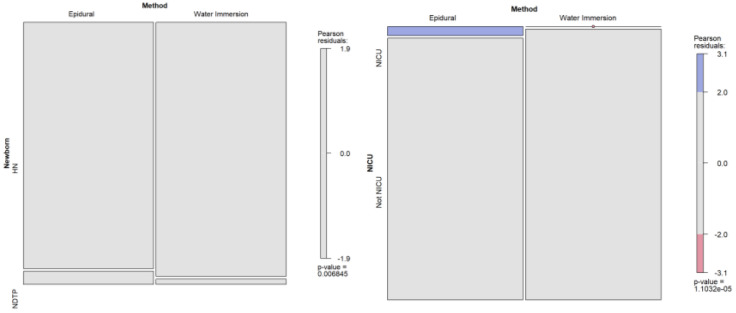
Standardized residuals for newborn outcomes among study groups. Notes: HNs = healthy newborns; NDTPs = newborns with difficulties in the transitional period; NICUs = newborns admitted to Neonatal Intensive Care Units; Not NICUs = newborns who did not require Neonatal Intensive Care Units.

**Table 1 healthcare-12-01919-t001:** Characteristics of women who participated in the study by groups (N = 1134).

Characteristic	Water Immersion Group (n = 567)	Epidural Group (n = 567)	*p* Value ^a^
**Women’s parity, n (%)**			**0.984**
Nulliparous	216 (38.1)	222 (39.2)	
Para 1	254 (44.8)	249 (43.9)	
Para 2	80 (14.1)	81 (14.3)	
Para 3	14 (2.5)	13 (2.3)	
Para 4	3 (0.5)	2 (0.3)	
**Age, n (%)**			**<0.001**
25 years old or less	110 (19.4)	149 (26.3)	
26–30 years old	172 (30.3)	183 (32.3)	
31–35 years old	186 (32.8)	150 (26.4)	
36 years old or more	99 (17.5)	85 (15)	
**Newborns’ weigh (in grams), M (SD)**	3387.11 (398.502)	3432.00 (455.777)	**<0.001**

^a^ The statistics were derived from exact chi-square test for parity and mothers’ age and Mann–Whitney U test for newborns’ weight.

**Table 2 healthcare-12-01919-t002:** Maternal outcomes of women who participated in the study among groups (N = 1134).

Maternal Outcomes	Water Immersion Group (n = 567)	Epidural Group (n = 567)	*p* Value ^a^
Delivery, n (%) ^b^	n = 567	n = 563	<0.001
Spontaneous vaginal birth	536 (94.5)	285 (50.6)	
Operative vaginal birth	27 (4.8)	155 (27.5)	
Cesarean section	4 (0.7)	123 (21.9)	
**Perineal integrity, n (%) ^c^**	n = 563	n = 440	<0.001
Intact	173 (30.7)	64 (14.6)	
Minor laceration ^d^	365 (64.8)	257 (58.4)	
Major laceration ^d^	2 (0.4)	4 (0.9)	
Episiotomy	23 (4.1)	115 (26.1)	

^a^ The statistics were derived from exact chi-square tests for categorical outcomes. ^b^ The analyses were based on women participants with available data. The number of participants with available delivery data in the epidural group was n = 563. ^c^ Women who had a cesarean birth were excluded from the analyses on perineal integrity, leaving n = 563 and 440 for the water immersion and epidural groups, respectively. ^d^ Minor lacerations were defined as first- or second-degree perineal lacerations. Major lacerations were defined as third- or fourth-degree perineal lacerations.

**Table 3 healthcare-12-01919-t003:** Outcomes of the newborns who participated in the study by groups (N = 1134).

Newborn Outcomes	Water Immersion Group (n = 567)	Epidural Group (n = 567)	*p* Value ^a^
**5 min Apgar score, mean (SD) ^b^ **	9.92 (0.789) (n = 567)	9.68 (0.789) (n = 566)	<0.001
**Umbilical cord arterial pH, mean (SD) ^c^**	7.273 (0.929) (n = 558)	7.249 (0.880) (n = 549)	<0.001
**Newborns, n (%) ^d^**	n = 567	n = 567	<0.001
HNs	555 (97.9)	538 (94.9)	
NDTPs	12 (2.1)	29 (5.1)	
**NICU admissions, n (%)**	n = 567	n = 567	<0.001
No	567 (100)	548 (96.6)	
Yes	0 (0)	19 (3.4)	

Notes: HNs = healthy newborns; NDTPs = newborns with difficulties in the transitional period; NICU = Neonatal Intensive Care Unit. ^a^ The statistics were derived from and Mann–Whitney U test for 5 min Apgar score and umbilical cord arterial pH and exact chi-square tests for categorical outcomes (pathological newborn and NICU admission). ^b^ The analyses were based on women included in the study with available data. The number of women included in the study with available delivery data in the epidural group was n = 566. ^c^ The analyses were based on women included in the study with available data. The number of women included in the study with available delivery data in the water immersion group was n = 558 and in the epidural group was n = 549. ^d^ Newborns with difficulties in the transitional period were defined as those who presented an Apgar score < 7 at 5 min of life and/or a postpartum umbilical cord pH < 7.10.

## Data Availability

Data are available on request from the corresponding author.

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
