# Peer review of "Effects of Water Immersion Versus Epidural as Analgesic Methods during Labor among Low-Risk Women: A 10-Year Retrospective Cohort Study"

_healthcare, 2024, doi:10.3390/healthcare12191919_

Round 1

Reviewer 1 Report

Comments and Suggestions for Authors

1- The title needs to be short and concise.

2- introduction:

A-What are the potential long-term effects of water immersion during labor on both the mother and the newborn?

B-Are there any specific guidelines or criteria for determining which women are suitable candidates for water immersion during labor?

C-How do the costs of water immersion compared to epidural analgesia, and what are the potential economic implications of promoting water immersion as an analgesic option for childbirth?

3- method section :

Did the study address potential limitations or confounding factors that could have influenced the results?

4- Discussion part:

- What were the reasons behind the differences in the rates of spontaneous vaginal birth (SVB) and cesarean births between the groups using different analgesic methods?

- Did the study address the medium/long-term results on the pelvic floor in relation to the use of different analgesic methods?

Author Response

Comment 1: The title needs to be short and concise.

Response 1: Thank you for your suggestion. The tittle has been modified to comply to your suggestion, maintaining the format that the journal requests.

Comment 2: What are the potential long-term effects of water immersion during labor on both the mother and the newborn?

Response 2: Thank you for your valuable question. We can only try to hypothesize what the long-term beneficial effects of using water immersion versus epidural analgesia as a method of pain relief in childbirth might be, since the objective of this study was to compare the immediate obstetric effects (mode of birth and perineal integrity) and short-term neonatal effects of using both analgesic techniques.

Based on the results obtained in our study (reduction in episiotomy, caesarean section and OVB rates; as well as increase in intact perineum rate), we believe that there could be a number of potential benefits from the use of water immersion in the long term. According to several systematic reviews, an unjustified high incidence of caesarean births leads to increased maternal and fetal morbidity. This would include the possibility of the late development of multiple morbidities in both children and adults resulting from the gut flora of neonates being deprived of normal colonisation by maternal vulvovaginal and rectal flora, allowing it to be colonised by ‘bad’ bacteria that may compromise the child's immune system. This may explain its association with obesity at an early age, food allergy, asthma, type 1 diabetes mellitus and various forms of dermatitis1-3. There has also been found to be a negative association with early breastfeeding with consequent implications for infant wellbeing4 . Finally, caesarean birth also determines the presence of risk factors for fetal/neonatal morbidity in subsequent pregnancies such as preterm birth, unexplained fetal death, placenta previa, invasively adherent placenta and even impaired fetal head and increased risk of stillbirth5-7. A high incidence of operative vaginal births (OVB) also appears to produce a number of adverse effects, including an increased incidence of occult anal sphincter injury8, late urinary and anal incontinence, as well as neonatal convulsions, intracranial haemorrhage and jaundice9. On the other hand, it has been shown that selective episiotomy policies result in fewer women with severe perineal/vaginal trauma and its possible subsequent consequences10.  Especially the restrictive use of mediolateral episiotomy could reduce the incidence of sexual dysfunction, with decreased libido, decreased ability to achieve orgasm, decreased lubrication, disruption of the vaginal epithelium and the development of vaginal atrophy that can cause dyspareunia11.

In order to better relate the answer to your question to the results obtained in our study, we believe it is more valuable to summarise these potential long-term beneficial effects in the Discussion section (thus linking it to the answer to your 4B question).

1. Carrapato MRG, Ferreira AM, Wataganara T. Cesarean section: the pediatricians' views.J Matern Fetal Neonatal Med. 2017;30(17):2081-2085. doi: 10.1080/14767058.2016.1237496.

2. Kulas T, Bursac D, Zegarac Z, Planinic-Rados G, Hrgovic Z. New Views on Cesarean Section, its Possible Complications and Long-Term Consequences for Children's Health. Med Arch. 2013;67(6):460-3. doi: 10.5455/medarh.2013.67.460-463.

3. Boutsikou T, Malamitsi-Puchner A. Caesarean section: impact on mother and child. Acta Paediatr. 2011;100(12):1518-22. doi: 10.1111/j.1651-2227.2011.02477.x.

4. Prior E, Santhakumaran S, Gale C, Philipps LH, Modi N, Hyde MJ. Breastfeeding after cesarean delivery: a systematic review and meta-analysis of world literature. Am J Clin Nutr. 2012;95(5):1113-35. doi: 10.3945/ajcn.111.030254.

5. Bahl R, Hotton E, Crofts J, Draycott T. Assisted vaginal birth in 21st century: current practice and new innovations. Am J Obstet Gynecol. 2024;230(3S):S917-S931. doi: 10.1016/j.ajog.2022.12.305.

6. Aabakke AJ, Krebs L, Lykke JA. [Caesarean section may have long-term consequences for both mother and child].  Ugeskr Laeger. 2014 Apr 22;176(17):V12130729.

7. Timor-Tritsch IE, Monteagudo A. Unforeseen consequences of the increasing rate of cesarean deliveries: early placenta accreta and cesarean scar pregnancy. A review. Am J Obstet Gynecol. 2012;207(1):14-29. doi: 10.1016/j.ajog.2012.03.007.

8. Johnson JK, Lindow SW, Duthie GS. The prevalence of occult obstetric anal sphincter injury following childbirth--literature review. J Matern Fetal Neonatal Med. 2007;20(7):547-54. doi: 10.1080/14767050701412917.

9. Vayssière C, Beucher G, Dupuis O, Feraud O, Simon-Toulza C, Sentilhes L, et al. French College of Gynaecologists and Obstetricians. Instrumental delivery: clinical practice guidelines from the French College of Gynaecologists and Obstetricians.   Eur J Obstet Gynecol Reprod Biol. 2011;159(1):43-8. doi: 10.1016/j.ejogrb.2011.06.043.

10. Jiang H, Qian X, Carroli G, Garner P. Selective versus routine use of episiotomy for vaginal birth. Cochrane Database Syst Rev. 2017 Feb 8;2(2):CD000081. doi: 10.1002/14651858.CD000081.pub3.

11. Okuyan E, Akgül ÖK, Günakan E. Mediolateral Episiotomy in Nulliparous Women Increases the Risk of Sexual Dysfunction. Bagcilar Medical Bulletin, 2023;8(1), 21-26. doi:10.4274/BMB.galenos.2022-11-091

Comment 3: Are there any specific guidelines or criteria for determining which women are suitable candidates for water immersion during labor?

Response 3: There are specific guidelines that contemplate the inclusion criteria that women must meet to be considered candidates for immersion in water. Two of them are cited in the bibliographic references of the manuscript, one from the Department of Health (Western Australia) from 2017 (Reference 6, page 4) and another one from the National Health System from 2022 (Reference 7, section 3.1.)

Comment 4: How do the costs of water immersion compared to epidural analgesia, and what are the potential economic implications of promoting water immersion as an analgesic option for childbirth?+

Response 4: Thank you very much for your question, as we consider comparative cost analysis to be a topic of particular interest for policy decisions regarding the implementation of a change in standard practice such as the generalisation of this type of analgesic measure (water inmersion -WI-). Although WI has high initial costs due to the need to set up the necessary spaces and infrastructure (bathtub, etc.), in the long term it can lead to significant savings. There could be a reduction in direct healthcare costs due to the reduced use of analgesic medication and associated procedures (establishment and maintenance of a venous line and the insertion and maintenance of an epidural catheter, etc.), as well as a reduction in associated complications (lower rate of caesarean sections, perineal sutures due to tearing, episiotomies, lower rate of NICU admissions, etc.).

In fact, a cost analysis based on the results obtained in the retrospective cohort study presented here is currently underway to study its cost-effectiveness.

Comment 5: Did the study address potential limitations or confounding factors that could have influenced the results?

Response 5: 

Unfortunately, there are several confounding variables that could have affected the results of the study. For example, the influence of physical activity during pregnancy or the practice of antenatal perineal massage. These variables, which may affect on various obstetric and/or neonatal outcomes, were not measured and, therefore, their effect on the results found, could not be evaluated. This is, therefore, an important limitation of the study and, as such, we have reflected it in the Study limitations section.

Specifically, the following information has been added:

“Finally, there are several confounding variables that could have affected the results of the study. For example, the influence of physical activity during pregnancy or the practice of antenatal perineal massage. In this sense, Michaleck et al. report that physical activity is safe during pregnancy, but that there are no significant differences in birth outcomes. 48  Only Domenjoz et al. reported a lower risk of caesarean birth in women who carried out a structured program of physical activity during pregnancy.  49  On the other hand, Abdelhakim et al. reported that antenatal perineal massage reduces the risk of severe tears and episiotomies. 50  These variables, which may affect on various obstetric and/or neonatal outcomes, were not measured, as there was not a specific registry on these main outcomes, and therefore this retrospective information could not be gathered.  However, we would like to notice that all women equally received advice about physical activity during pregnancy and antepartum perineal massage.”

Comment 6: What were the reasons behind the differences in the rates of spontaneous vaginal birth (SVB) and cesarean births between the groups using different analgesic methods?

Response 6: Again, we can only attempt to establish hypotheses to justify the differences found, as the design of our study does not allow to establish causal relationships. The scientific literature shows that the use of epidural analgesia (EA) results in a prolongation of labour stages and an increase in assisted vaginal births rate, although it has not been shown to result in a higher rate of caesarean section compared to placebo or an untreated group1. However, in some studies, consistent with our results, the use of WI has been found to increase the incidence of spontaneous vaginal births (SVB) as well as a reduction in caesarean births compared to conventional deliveries. Perhaps SVB is facilitated with WI because buoyancy and ease of movement make it easier for women to maximise pelvic diameters, which may lead to better fetal flexion and easier delivery. Also warm water increases maternal relaxation reducing pain perception, and may lead to improved uterine perfusion and reduced2. Also, since WI can alleviate labor pain and provide more personalized services for the mother during labor, it may reduce the number of cesarean deliveries performed for social factors, thereby reducing the unnecessary cesarean sections and thereby caesarean section rate3.

1. Anim-Somuah M, Smyth RMD, Cyna AM, Cuthbert A. Epidural versus non-epidural or no analgesia for pain management in labour. Cochrane Database Syst Rev. 2018;5(5):         CD000331.. doi:10.1002/14651858.CD000331.pub4

2.Henderson J, Burns EE, Regalia AL, Casarico G, Boulton MG, Smith LA. Labouring women who used a birthing pool in obstetric units in Italy: prospective observational study. BMC Pregnancy Childbirth. 2014;14:17. Published 2014 Jan 14. doi:10.1186/1471-2393-14-17

3. Liu Y, Liu Y, Huang X, et al. A comparison of maternal and neonatal outcomes between water immersion during labor and conventional labor and delivery. BMC Pregnancy Childbirth. 2014;14(1):160.. doi:10.1186/1471-2393-14-160

Comment 7: Did the study address the medium/long-term results on the pelvic floor in relation to the use of different analgesic methods?

Response 7: This issue has been addressed in response to comment 2

Reviewer 2 Report

Comments and Suggestions for Authors

Dear authors; I read your article with pleasure. I saw that it also has positive effects on the pelvic floor. Today, sexual dysfunctions and aesthetic concerns as a result of unnecessary episiotomies are increasingly making women's lives difficult and more and more patients are applying to clinics for solutions in this regard. I have been working in pelvic floor and cosmetic gynecology as a special interest for a long time. For this reason, adding a sentence with the following reference to your study, including its positive effects on episiotomy results and indirectly contributing to the protection of women's sexual health, will make your study more compact and more valuable. 

Ref: Okuyan, E., Akgül, Ö.K., Günakan, E. (2023). Mediolateral Episiotomy in Nulliparous Women Increases the Risk of Sexual Dysfunction. Bagcilar Medical Bulletin, 8(1), 21-26. doi:10.4274/BMB.galenos.2022-11-091.

Author Response

Comment 1:  I read your article with pleasure. I saw that it also has positive effects on the pelvic floor. Today, sexual dysfunctions and aesthetic concerns as a result of unnecessary episiotomies are increasingly making women's lives difficult and more and more patients are applying to clinics for solutions in this regard. I have been working in pelvic floor and cosmetic gynecology as a special interest for a long time. For this reason, adding a sentence with the following reference to your study, including its positive effects on episiotomy results and indirectly contributing to the protection of women's sexual health, will make your study more compact and more valuable. 

 Ref: Okuyan, E., Akgül, Ö.K., Günakan, E. (2023). Mediolateral Episiotomy in Nulliparous Women Increases the Risk of Sexual Dysfunction. Bagcilar Medical Bulletin, 8(1), 21-26. 

Response 1: Thank you for your positive evaluation. We appreciate your feedback. We have included your valuable suggestion about the relationship between episiotomy (MLE) and sexual dysfunctions in the Discussion section (including the proposed bibliographical reference).

Reviewer 3 Report

Comments and Suggestions for Authors

Summary and overall impression

The manuscript with the title “The effect of water immersion and epidural as analgesic methods during labor in maternal and perinatal outcomes among low-risk women attended in a first level hospital: A 10 years retrospective cohort study“ by Herrero-Orenga et al. presents a single-center, retrospective analysis of 1134 childbirths using either water immersion or epidural anesthesia for pain management. 

The authors address a topic of high interest and obstetric relevance. By encompassing a large cohort of women for both groups, the study adds evidence to the literature on this topic. The structure and language of this manuscript is of high quality, method description and data presentation are mostly adequate, and it is well and concisely written. For improvement of this work, only minor revision is considered necessary. 

Discussion of issues for improvement:

GENERAL ISSUES:

1.     There are some typographical errors (e.g., table 1) and minor grammar issues. Please revise the manuscript thoroughly (e.g., line 81, 87, 123). Furthermore, please be consistent in using either ‘caesarean’ or ‘cesarean’ section (see line 280, 283). In the abstract, the word order in ‘neonatal intensive care unit’ should be corrected. 

2.     The authors should reconsider the use of the expression ‘pathological’ and ‘non-pathological’ newborns as it strongly implies fetal malformations and sounds irritating to the reader. In the way the expression is used, other descriptions might be beneficial.

3.     Please be sure to provide all abbreviations used in figures and tables in the corresponding caption.

INTRODUCTION

1.     Line 62 and following: It is preferable to simple use the expression ‘Apgar score’ instead of ‘Apgar test score’.

2.     Line 65 and 81: It seems that words are missing for the correct sense of the sentence. 

METHODS

1.     Line 132, 185 etc: It is advisable, to rephrase the expression ‘women participated in the study’ as it is no active decision rather than a retrospective inclusion in the analysis. 

2.     Participants, line 134-135: As it is not quite clear: Did water immersion remain the only method of pain relief throughout the course of labor and childbirth or has there been the need for additional methods in this group? 

3.     Participants, line 137-140: It is necessary to provide information on how the subjects with EA were selected from the 4445 eligible women in this time period. 

RESULTS

1.     Please check thoroughly the congruence of numbers in full text and corresponding tables.

2.     Line 187: What is the unit of the number in brackets following the mean age?

3.     Table 1: The expressions ‘Secondiparous, etc…’ sound rather unusual. The authors may think of another way to depict degree of parity.

4.     Table 3: How come there is no delivery data for all subjects in the EA group, when these cases could have been chosen from 4445 eligible women (also when in line 365 there can be found a statement that no unfilled fields were detected in the extracted data?

5.     Figure 2, Delivery Outcome: The aspects of ‘eutopic’ and ‘dystopic labor’ have not been explained in the full text- or is it simply the figure labeling which should be adapted?

6.     Table 2 and 3: It is recommendable to depict the actual correct n for every corresponding line/column when it differs from n=567.

DISCUSSION

1.     Discussion/Study limitation: It is remarkable, that the difference in mode of delivery in the present work (SVB rate: 94.5 vs 50.6%) is greater than the percentages in the cited literature (see line 279-292). Therefore, it is inevitable to explain and discuss the selection of ‘control’ subjects in the EA group, as this might be a significant bias for analysis. For the reader of the study, it is not transparent, how the authors selected the EA cases (see aspect nr. 2, methods).

Comments on the Quality of English Language

Please refer to the above mentioned comments. 

Author Response

Comment 1: The manuscript with the title “The effect of water immersion and epidural as analgesic methods during labor in maternal and perinatal outcomes among low-risk women attended in a first level hospital: A 10 years retrospective cohort study“ by Herrero-Orenga et al. presents a single-center, retrospective analysis of 1134 childbirths using either water immersion or epidural anesthesia for pain management. 

The authors address a topic of high interest and obstetric relevance. By encompassing a large cohort of women for both groups, the study adds evidence to the literature on this topic. The structure and language of this manuscript is of high quality, method description and data presentation are mostly adequate, and it is well and concisely written. For improvement of this work, only minor revision is considered necessary.

Response 1: Thank you for your positive evaluation and recognition of our paper’s potential contribution to the literature.

Comment 2: There are some typographical errors (e.g., table 1) and minor grammar issues. Please revise the manuscript thoroughly (e.g., line 81, 87, 123). Furthermore, please be consistent in using either ‘caesarean’ or ‘cesarean’ section (see line 280, 283). In the abstract, the word order in ‘neonatal intensive care unit’ should be corrected. 

Response 2: Thank you for the suggestion. We have reviewed the manuscript and corrected the grammar issues. We finally have used the term caesarean section or caesarean birth throughout the manuscript. 

Comment 3: The authors should reconsider the use of the expression ‘pathological’ and ‘non-pathological’ newborns as it strongly implies fetal malformations and sounds irritating to the reader. In the way the expression is used, other descriptions might be beneficial.

Responde 3: According to the reviewer’s suggestion we have changed both ‘pathological’ and ‘non pathological’ newborns expressions. We have renamed them as ‘healthy newborns’ (HN) for non-pathological newborns, and ‘newborns with difficulties in the transitional period’ (NDTP) for pathological newborns. We have corrected these expressions along the manuscript and in tables and figures. 

Comment 4: Please be sure to provide all abbreviations used in figures and tables in the corresponding caption.

Response 4: According to the reviewer’s suggestion, we have added the missing abbreviations in figures and tables. 

Comment 5: Line 62 and following: It is preferable to simple use the expression ‘Apgar score’ instead of ‘Apgar test score’.

Response 5: Thank you for the suggestion. We have proceeded to modify the cited expression in the manuscript.

Comment 6: Line 65 and 81: It seems that words are missing for the correct sense of the sentence. 

Response 6: Thank you for the observation. We have proceeded to modify the missing words in the manuscript.

Comment 7: Line 132, 185 etc: It is advisable, to rephrase the expression ‘women participated in the study’ as it is no active decision rather than a retrospective inclusion in the analysis. 

Response 7: We appreciate your contribution and we have proceeded to modify it in the manuscript.

Comment 8: Participants, line 134-135: As it is not quite clear: Did water immersion remain the only method of pain relief throughout the course of labor and childbirth or has there been the need for additional methods in this group? 

Response 8: For ethical reasons, women in the WI cohort were given stronger analgesics when they required it. This situation occurred only in 30 women out of the 567 in the WI cohort.

Comment 9: Participants, line 137-140: It is necessary to provide information on how the subjects with EA were selected from the 4445 eligible women in this time period. 

Response 9: More information on how the subjects in the EA group were selected has been provided.

Specifically, we have added the following information in the Participants subsection:

“567 women were included in the study as the WI group. They were all the women that chose the immersion in water as analgesic method during childbirth from June 1st 2009 to May 31st 2019 and began WI after admission. 

Regarding the EA group, 4445 women chose epidural as the analgesic method during childbirth, despite meeting the conditions to use WI, during the same time period, and met the inclusion criteria. Out of these 4445 eligible participants, 567 women were selected following a stratified random sample 1:1. The sample was stratified by parity. To randomize the sample the Excel formula (RANDOM) was used to assign a random number to each row of the list. Then, the number of records needed for the study were indicated (the same as the number of women who had used water immersion in each sample stratum). 

In all, 1134 women, who met the clinical criteria, were included in the study: 567 women for the WI group and 567 women for the EA group.”

Comment 10: Please check thoroughly the congruence of numbers in full text and corresponding tables

Response 10: Thank you for the suggestion. We have checked the congruence of numbers in full text and corresponding tables.

Comment 11: Line 187: What is the unit of the number in brackets following the mean age?

Response 11: The number refers to the standard deviation or standard error. This information has been now included in the text.

Comment 12: Table 1: The expressions ‘Secondiparous, etc…’ sound rather unusual. The authors may think of another way to depict degree of parity.

Response 12: Thank you for the suggestions. We have used a more common way to describe women parity at the table.

Comment 13: Table 3: How come there is no delivery data for all subjects in the EA group, when these cases could have been chosen from 4445 eligible women (also when in line 365 there can be found a statement that no unfilled fields were detected in the extracted data)?

Response 13: Delivery data are included in Table 2. Only data from the randomly selected women in the EA group were gathered and, therefore, included for the analyses. For more details of the randomization process, please see our answer to the second point made in the Methods section.

Regarding line 365, this was a mistake by our side. We have rewritten the sentence and we believe it is easier to understand now. The sentence is as follows:

Firstly, this is a retrospective study and therefore the data recording was carried out in the computerized clinical history by different health care providers. However, a negligible number of missing data were found in the extracted data. Moreover, the sample size was large and obtained over a period of 10 years.

Comment 14: Figure 2, Delivery Outcome: The aspects of ‘eutopic’ and ‘dystopic labor’ have not been explained in the full text- or is it simply the figure labeling which should be adapted?

Response 14: According to the reviewer’s suggestion, labeling of ‘eutocic’ and ‘dystocic’ has been adapted in Figure 2. Specifically, we have changed the terms as follows:

Spontaneous vaginal Birth (SVB) for eutocic labor.

Operative Vaginal Birth (OVB) for dystocic labor

Caesarean section (CS) for caesarean or cesarean section.

Comment 15: Table 2 and 3: It is recommendable to depict the actual correct n for every corresponding line/column when it differs from n=567.

Response 15: Considering the reviewer’s recommendation, n for every analysis and subgroup has been identified in each table. 

Comment 16: Discussion/Study limitation: It is remarkable, that the difference in mode of delivery in the present work (SVB rate: 94.5 vs 50.6%) is greater than the percentages in the cited literature (see line 279-292). Therefore, it is inevitable to explain and discuss the selection of ‘control’ subjects in the EA group, as this might be a significant bias for analysis. For the reader of the study, it is not transparent, how the authors selected the EA cases (see aspect nr. 2, methods)

Response 16: We have already resolved this issue in comment 8